# Polymer-Based Nanostructures for Pancreatic Beta-Cell Imaging and Non-Invasive Treatment of Diabetes

**DOI:** 10.3390/pharmaceutics15041215

**Published:** 2023-04-11

**Authors:** Shakila Behzadifar, Alexandre Barras, Valérie Plaisance, Valérie Pawlowski, Sabine Szunerits, Amar Abderrahmani, Rabah Boukherroub

**Affiliations:** Univ. Lille, CNRS, Centrale Lille, Univ. Polytechnique Hauts-de-France, UMR 8520, IEMN, F-59000 Lille, France; shakila.behzadifar@univ-lille.fr (S.B.); alexandre.barras@univ-lille.fr (A.B.); amar.abderrahmani@univ-lille.fr (A.A.)

**Keywords:** diabetes, beta cells mass, polymer-based nanostructures, imaging, antidiabetic drug delivery, foot ulcers

## Abstract

Diabetes poses major economic, social, and public health challenges in all countries worldwide. Besides cardiovascular disease and microangiopathy, diabetes is a leading cause of foot ulcers and lower limb amputations. With the continued rise of diabetes prevalence, it is expected that the future burden of diabetes complications, early mortality, and disabilities will increase. The diabetes epidemic is partly caused by the current lack of clinical imaging diagnostic tools, the timely monitoring of insulin secretion and insulin-expressing cell mass (beta (β)-cells), and the lack of patients’ adherence to treatment, because some drugs are not tolerated or invasively administrated. In addition to this, there is a lack of efficient topical treatment capable of stopping the progression of disabilities, in particular for treating foot ulcers. In this context, polymer-based nanostructures garnered significant interest due to their tunable physicochemical characteristics, rich diversity, and biocompatibility. This review article emphasizes the last advances and discusses the prospects in the use of polymeric materials as nanocarriers for β-cell imaging and non-invasive drug delivery of insulin and antidiabetic drugs in the management of blood glucose and foot ulcers.

## 1. Introduction

Diabetes is a pandemic non-communicable disease affecting all countries worldwide. The prediction regarding the global prevalence of the disease in the next years is highly alarming, according to the International Diabetes Federation, due to its steady rise in numbers. While diabetes affected 366 million people in 2011, the disease is expected to reach 552 million by 2030 [1]. The World Health Organization (WHO) differentiates between four types of diabetes according to etiopathogenesis. Besides gestational diabetes (GD) and other forms of diabetes that gather iatrogenic diabetes, exocrine pancreatic diseases-induced diabetes (such as cystic fibrosis) and virus-induced diabetes [2], type 1 diabetes (T1D), and type 2 diabetes (T2D) are the most prevalent forms. Diabetes reduces life expectancy on average by 6 years, mainly caused by cardiovascular and cerebrovascular diseases [1]. In high-income countries, diabetes ranks fourth or fifth as the leading cause of death. Since the last decade, the disease has become a major risk factor for early mortality in middle- and low-income countries [3]. Diabetes also can lead to disabilities, including reduced mobility caused by foot ulcers [4,5]. Up to one-third of diabetic patients may suffer from diabetic foot ulcers during their life [6]. This complication is challenging as it frequently becomes infected. Therefore, this causes hospital admissions and more than half of non-traumatic lower limb amputations. The medical and economic burden of diabetic foot ulcers is huge. It has been estimated that the cost ascribed to diabetic foot ulcers is as high as $45,000 per patient [7]. In reality, the cost is underestimated, as it should include the psychosocial impact on the patient’s quality of life caused by impaired mobility and substantial loss of productivity. The treatment results of diabetic foot ulcers are often unsatisfactory, especially in advanced cases. Moreover, the recurrence rate is high despite the healing of the ulcer. Currently, treatments available for diabetic foot ulcers include debridement, pressure relief, antibiotics, and revascularization [8]. In many cases, these therapies fail to ensure full wound repair [9,10]. Therefore, there is a need to find out innovative topical therapies for promoting tissue regeneration while minimizing complications in the wound area.

Diabetic foot ulcers and other complications ensue when diabetes diagnosis is late and when glycemia level is uncontrolled despite the treatment. In T1D, hyperglycemia results from massive destruction by autoimmunity of pancreatic insulin-producing cells called “β-cells” during childhood or adolescence. Patients with T1D usually lose more than 80% of the β-cell mass, and, therefore, they do not have sufficient plasma insulin levels to regulate their blood glucose [11]. The main treatment of T1D consists of changing the lifestyle of patients together with insulin injections using prefilled pens. However, the fear of needles and the pain of injection make insulin treatment uncomfortable, although some progress has been made in the development of painless needles [12]. Consequently, the adherence of patients to the treatment reduces, leading to neglecting or stopping the treatment in some cases [13]. Replacing prefilled pens with non-invasive routes of insulin delivery is currently a key challenge. Such alternatives would enable not only the achievement of better glycemic control but also slow the development of foot ulcers and other complications. For a few patients with T1D, insulin therapy could be replaced by islet cell transplantation [14]. However, the procedure can fail as the transplanted β-cells can be destructed by patient immune cells and other mechanisms [15]. At the present time, the mechanism of β-cell failure cannot be directly determined. It is often predicted by uncontrolled plasma glucose levels [16].

The absence of medical imaging technologies, allowing the monitoring of β-cell mass (BCM) and function, slows clinicians from taking adequate actions for patients. The lack of imaging methods is also a concern for the management of glycemia in T2D. The disease is the most prominent type of diabetes, with 80–85% of all diagnosed cases [17]. T2D develops when insulin production by β-cells cannot cope with the insulin body demand provoked by insulin resistance [17,18,19,20]. Defective insulin secretion, due to β-cells dysfunction, is the primary cause of the reduced insulin plasma levels [21,22,23]. With β-cells dysfunction, there is a loss of BCM, which contributes to the diminution of insulin production. It has been reported that the BCM of patients with T2D is 40–60% of that of a body mass index-matched non-diabetic person [23,24]. In T2D, like in T1D, the evaluation of BCM is not estimated by the time of diabetes diagnosis. Therefore, it happens that the medicine is inappropriately prescribed, in particular, if the BCM is reduced. In addition, as none of the current antidiabetic drugs can prevent or stop the reduction of BCM, for many patients, insulin injection with prefilled pens will become the only therapeutic option, like in T1D, with the problems of adherence as discussed above. Therefore, along with BCM imaging, the development of non-invasive modes for delivering insulin is also a requirement for managing glycemia and thereby reducing the complication risks in T2D.

In medical applications, nanoparticles (NPs) are becoming increasingly popular because of their unique and multiple properties. High surface-to-mass ratios, quantum confinement properties, and the ability to adsorb high amounts of active principles are some of the features that distinguish them. In the last years, polymers have created major innovations in the diagnosis and therapy of some diseases [25,26,27]. A combination of polymer science and nanotechnology has contributed to new developments in drug delivery, food preservation, packaging, tissue engineering, nano-implants etc. A number of polymeric nanostructured materials (PNMs) have been used to diagnose and treat diseases [28,29,30]. The PNMs include micelles, polymersomes, nanoparticles, nanocapsules, nano-gels, nanofibers, dendrimers, brush polymers, and nanocomposites. Size, shape, surface charge, surface chemistry, mechanical strength, porosity, and other properties of these materials can be adjusted to meet the specific functionalities of the targeted biomedical application [31]. Polymeric nanoparticles (PNPs) are colloidal particles with sizes ranging from 1 to 100 nanometers. PNPs refer to any type of polymer nanoparticles, specifically nanospheres and nanocapsules. In addition to their high surface area-to-volume ratio, PNPs can be used for imaging, sensing, and drug delivery. In order to increase their specificity and efficacy, targeting ligands, imaging agents, and therapeutic payloads can be added [32,33]. Several water-soluble and insoluble drugs and biologically active compounds have been significantly boosted by PNPs for targeted drug delivery by improving their bioavailability, safety, and biocompatibility, therefore reducing toxicity and adjusting retention time [34]. PNPs are one of the most popular nanotechnological delivery vehicles. PNPs have been extensively studied for their potential ability to accumulate at disease sites, both passively, through enhanced permeability and retention (EPR) and actively. Therefore, PNPs have been successfully used in different therapeutic applications, and many more are in various stages of development [35,36,37].

Depending on their origin, polymers can be divided into synthetic and natural. Synthetic polymers can be classified into biodegradable and non-biodegradable groups, while natural polymers are mainly categorized into protein- and polysaccharide-based compounds. Figure 1 presents a classification of polymers based on their origin. Research has centered on biodegradable nanocomposites in recent years due to the properties of biodegradable polymer matrices, including sustained, controlled and delayed release, and controlled degradation. In addition, byproducts of biodegradable polymers are biocompatible or harmless after they undergo enzymatic or nonenzymatic biodegradation processes in the body [38,39,40,41]. Several disadvantages of non-biodegradable polymers have been well-documented, including their toxicity in the body upon accumulation. Clinical settings use a variety of non-biodegradable polymers, while scientists are more interested in developing biodegradable and biocompatible polymers for drug delivery systems [42].

Many review articles have been published on the application of polymers or nanotechnology for diabetes management. The topics covered in these articles range from the disease’s pathophysiology to its epidemiology and treatment [43,44,45,46,47]. Nevertheless, there is a lack of papers covering the two aspects of polymers and nanostructures in diabetes management. The objective of this review article is to discuss the latest advances in the use of polymer-based nanostructures in the imaging of β-cells mass and function using fluorescence and magnetic resonance imaging techniques as well as drug delivery through different approaches, such as oral, buccal, nasal, pulmonary, and transdermal routes. In addition, we summarize the progress being made in improving wound healing via these nanostructures. Considering the wide variety of audiences interested in diabetes, a review article on polymer nanostructures will be interesting for researchers, clinicians, and public health professionals. Figure 2 illustrates a summary of ways that polymer-based nanostructures can help in the management of diabetes.

## 2. Type 1 Diabetes (T1D) Treatment

Diabetes is caused by a lack of insulin, which controls blood glucose levels. As mentioned above, there are β-cells throughout the pancreas, which produce insulin. T1D is caused by the autoimmune destruction of β-cells. To control blood glucose levels in T1D, multiple daily insulin injections have been used. To summarize, diabetes patients typically take insulin to improve their condition; however, due to bioavailability issues, it is usually administered subcutaneously [48,49,50,51]. A century ago, insulin was discovered, and this groundbreaking discovery led to a Nobel Prize in Physiology or Medicine in 1923 to Frederick Grant Banting and Professor John James Richard Macleod [52]. Diabetes can be controlled most effectively and widely with insulin. In comparison to conventional drugs, peptides and proteins have proven to be very effective in biomedicine due to their high levels of activity, specificity and efficacy [53,54].

### 2.1. Delivery of Insulin via Polymeric Nanostructures

Drug delivery systems have recently been redesigned in a variety of novel ways. In addition to reducing degradation and associated side effects, increasing bioavailability, and facilitating drug accumulation in preferred biological sites, many of these approaches are also developed quickly to improve patient compliance [55,56,57]. As part of a patient’s treatment regimen, insulin should be administered by injection with hypodermic needles. Continuous glucose monitoring and frequent insulin injections, however, cause patients significant inconvenience. This results in pain, inflammation, infection risk, thickening of the skin, tissue necrosis, and nerve damage as a result of continuous glucose monitoring. A smart drug delivery system must be developed to overcome the limitations and drawbacks of open-loop delivery [58,59,60]. The number of articles published in the field of insulin drug delivery by oral administration between 2000 and 2023 indicates that this field has received considerable attention. By analyzing the keywords insulin and nanoparticle/nanostructure, it can be seen that nanotechnology was considered in the drug delivery of insulin, Figure 3.

There has been a significant increase in polymer science in recent decades, as well as improvements in drug delivery systems (DDSs) using polymer-based materials. Polymer nanoparticles (PNPs) can be modified for their physicochemical properties, including charge and association efficiency. They can be tailored to enhance insulin stability and bioavailability, control release profiles, stabilize systems and modulate biological behavior [61]. Polysaccharides, hyaluronic acid, chitosan, glycol chitosan, pullulan, and dextrin, are commonly used for drug delivery [62]. PNPs and nanoplatforms/carriers can be incorporated into an insulin DDS. Novel insulin formulations can be developed by combining these systems, with the aim of improving insulin bioavailability, half-life circulation, bioavailability, and release profiles. Nanoplatform selection is determined by the administration route, which affects pharmacokinetic properties. The form of drug carrier, formulation size, and core material of these nanoplatforms can be used to classify them. As seen in Figure 4, nanocarriers can be nanospheres and nanocapsules, dendrimers, and nanogels for insulin delivery [44,56,63,64]. In order to administer the drug, there are a number of different ways that can be used. Table 1 is a brief summary of these routes [50,57].

#### 2.1.1. Oral Delivery

For oral administration of proteins and peptides, the use of biocompatible and biodegradable nanoparticles has been suggested [65,66]. In particular, it has been claimed that PNPs are excellent candidates for oral delivery and the bioavailability of insulin. Oral insulin delivery is considered the most popular method due to its convenience, painlessness, and ease of self-administration. It is possible to adjust the insulin dosing schedule according to the efficacy and toxicity of the drug for each patient. As an additional benefit, oral insulin mimics the endogenous pathway of post-secretory insulin through a first pass through the liver instead of being absorbed into the bloodstream [67,68]. Oral administration of insulin also reduces allergic reactions, lipodystrophy around the injection site, and infectious disease transmission risks. In addition, the cost of oral insulin delivery for the healthcare system is not onerous when compared to insulin injections since it does not require specialized people for its administration and reduces hospital visits. In order to increase insulin bioavailability, several barriers need to be overcome along the gastrointestinal tract. There are several concerns about the activity and effectiveness of the protein, including acidic stomach conditions, active enzymes, mucus in the intestinal epithelium, and low permeability across the intestinal epithelium [69]. In this context, there is an added benefit in using copolymers such as *N*,*N*-dimethylaminoethyl methacrylate, polyanhydrides, polyurethanes, polyacrylic acids, and polyacrylamide for these applications, as they feature pH change-based swelling property that releases insulin in response to the change in pH [70,71]. *N*-isopropyl acrylamide, polyethylenimine, polymethacrylic acid, poly (isobutylcyanoacrylate), and poly(ε-caprolactone) are some of the materials used for oral insulin delivery [68].

Poly (glycolic acid) (PGA) is used in DDSs, owing to its favorable mechanical properties, compatibility with physiological conditions, and non-toxicity. The ester backbone of polyglycolide breaks down nonspecifically under physiological conditions. Due to its rapid degradation and insoluble nature in many common solvents, PGA-based DDSs are limited in their applications for research. Corn starch or sugarcane can be used to create PLA (polylactic acid). Indeed, the fermentation by bacteria of starch or sugars present in these plants allows the synthesis of lactic acid. The amino groups in PLA’s backbone make it more hydrophobic than PGA [72,73,74]. Poly (lactic-co-glycolic acid) or PLGA is a copolymer of PLA and PGA that provides exceptional bioavailability and sustained/controlled release properties. PLGA is a smart polymer known for its stimulus-sensitive behavior. As a smart polymer, PLGA is non-immunogenic, non-toxic, and biodegradable [75,76,77,78].

The potential use of D-α-tocopherol poly (ethylene glycol) 1000 succinate (TPGS)-emulsified poly (ethylene glycol) (PEG)-capped poly (lactic-co-glycolic acid) (PLGA) NPs for sustained insulin delivery was investigated. By using melt polycondensation, Malathi et al. synthesized low-molecular-weight polypropylene copolymers with PEG capping and TPGS emulsifier and investigated their efficacy as insulin delivery carriers [79]. Water–oil–water emulsion solvent evaporation was used to synthesize insulin-loaded TPGS-emulsified PEG-capped PLGA NPs. The NPs were spherical and had a loading efficiency of 78%. PEG-encapsulated PLGA NPs emulsion with TPGS prevented enzyme-induced degradation and aggregation in simulated GI fluids *in vitro*. As a result of insulin-loaded TPGS-emulsified PEG-capped PLGA NPs (ISTPPLG NPs), hypoglycemia was observed for 24 h. Oral administration of ISTPPLG6 NPs reduced blood glucose levels by up to 12 h when compared to oral administration of ISTPPLG4 NPs. Diabetic rats treated with ISTPPLG NPs had significantly lower cholesterol, urea, creatinine, and ALT levels. An illustration of the preparation of ISTPPLG NPs and an overview of the release of insulin from ISTPPLG NPs at different times is depicted in Figure 5. In conclusion, ISTPPLG6 NPs were found to be potential candidates for oral insulin delivery as a nanocarrier, owing to their regenerative effect on diabetic rats’ liver, kidney, and pancreas [79].

As a delivery system for patients with T1D, chitosan nanoparticles are considered the most promising. The hydrophilic nature of insulin prevents it from diffusing through the intestinal epithelium. Therefore, chitosan enhances insulin absorption by increasing permeation [70,80,81,82]. Chitosan is also water- and oxygen-permeable, and its hemostatic polymers can be blended or crosslinked to achieve physiologically significant release rates. The enzymes, chitosanase, papain, and lysozyme are responsible for the *in vitro* degradation of chitosan. The degradation rate depends on the chitosan polymer backbone’s degree of acylation and crystallinity [40,61,83]. A study from Zhang’s group has assessed the effect of cationic (poly(βcyclodextrins) (CPβCDs) on insulin release [84]. The polymer complex was formed by one-step polycondensation of alginate/chitosan nanoparticles using cyclodextrin (CD), epichlorohydrin (EP), and choline chloride (CC). The CPβCD NP may be effective in protecting insulin under simulated gastrointestinal conditions by forming complexes with insulin within the core of the alginate/chitosan nanoparticle, and insulin will be retained in this way. This is due to the polymeric chain and positive charge of CPβCDs. This nanoparticle had a mean size below 350 nm and could be loaded with insulin with an association efficiency (AE) of up to 87%. As a result of insulin retainment primarily within the core of the nanoparticles and good protection against degradation in simulated gastric fluid, cumulative insulin release in the simulated intestinal fluid was significantly higher (40% of total insulin release) compared to free insulin, i.e., without CPβCDs (18%). As a result of the preparation and release of nanoparticles, the insulin structure was also preserved. Figure 6 illustrates the oral insulin formulation of CPCDs-insulin-loaded alginate/chitosan nanospheres [84].

In the study of Cui et al., insulin-loaded nano-networks were prepared using a double-emulsion-based solvent evaporation method. To coat PLGA cores with oppositely charged nanoparticles, two natural polysaccharides, chitosan (positively charged) and alginate (negatively charged), were used as surfactants during the emulsion procedure to acquire oppositely charged nanoparticles. Therefore, a microgel containing insulin has been developed for long-term, sustained blood glucose control. The combination of microgel and nanocapsules produces a dynamic reservoir-like formulation that triggers the release of the drug when a stream of water (i.e., drug) builds up within the microgel. This strategy can also be used to deliver non-invasively and conveniently other therapeutics in addition to the mechanically triggered release. Figure 7 illustrates the formulation schematically [85].

#### 2.1.2. Buccal

The buccal mucosa is the inner lining of the cheeks and contains approximately 40–50 cell layers. Apart from its ease of access, this tissue has several other advantages, making it an ideal delivery site for drugs. It has been estimated that the buccal mucosa has the same level of permeability as the skin epithelium. It is highly vascularized, so absorbed materials reach the systemic circulation quickly. Acute cytotoxic effects are minimized by the rapid growth of cells, with complete turnover occurring once every 5–8 days. Even though hydrophilic drugs and proteins offer advantages, subcutaneous injections remain preferred. Permeation enhancers that loosen epithelial cell junctions and mucoadhesive polymers that hold compounds at high local concentrations for a long period of time are often used to increase trans-buccal drug delivery [57,86]. Lancina et al. used electrospun chitosan scaffolds for trans-buccal insulin delivery [87]. Different ratios of poly-ethylene oxide (PEO) were used for fiber morphology and physical properties control. In order to ensure insulin entrapped in fiber scaffolds remained bioactive during electrospinning, Akt-1 phosphorylation of pre-adipocytes exposed to chitosan fiber scaffolds was quantified. The buccal permeability of each fiber blend was determined in an *ex-vivo* porcine model. In degradation studies, PEO dissolves rapidly under normal conditions, leaving behind remodeled chitosan fibers. Upon spinning with chitosan:PEO at different ratios, the fibers were smaller, and insulin was released more rapidly. The bioactivity of insulin released from electrospun fiber mats was not impaired. It has been shown that the buccal mucosa permeability coefficient was approximately 500-fold higher for CS:PEO20 fibers compared to the other fiber blends and 16-fold higher for naked insulin, based on the steady-state flux region of the tests. It appears that electrospun chitosan nanofibers may be a viable vehicle for trans-buccal insulin delivery based on the findings of this study [87].

#### 2.1.3. Nasal Delivery

Nasal mucosal insulin administration could be an alternative method for treating diabetes. Using this strategy, insulin could be protected from extensive first-pass metabolism or digestive enzyme degradation. However, there are enormous challenges in this mode of administration, even though the nasal route can be patient-friendly for the circulatory system. Because of their hydrophilicity and large size, proteins have a smaller transfer scope than small hydrophobic molecules. Moreover, metabolic enzymes in the mammalian mucosa also contribute to the limitations of protein delivery [88,89].

The molecular properties of polymers containing boronic acid have gained significant scientific attention. In addition to generating reversible covalent borate esters with 1,2- or 1,3-diols, boronic acid-based polymers exhibit potential for saccharide sensing and controlled release [90,91]. Wei and his colleagues developed boronic acid-decorated dextran as vehicle matrices for the nasal delivery of insulin [92]. As compared with free insulin, phenylboronic acid-functionalized dextran nanoplatforms with excellent loading capacity exhibited significant endocytosis. The mechanism involved clathrin (a protein that plays a major role in the formation of coated vesicles) and lipid raft/caveolae-dependent endocytosis pathways. Combining clathrin-mediated endocytosis with lipid raft/caveolae-mediated endocytosis allowed the internalization of these dextran nanoplatforms. In addition, the nanoplatform-based phenylboronic acid-decorated polymer reduced blood sugar levels in rats by enhancing insulin uptake efficiency across epithelia. According to *in vivo* tests, these nanoplatforms did not cause nasal epithelial inflammation. They reduced blood sugar levels, improved insulin bioavailability, and demonstrated excellent mucoadhesive properties toward mucosal epithelial cells and high uptake rates by cells. These findings provided a promising approach for the development of diabetes therapeutic strategies using novel carriers of phenylboronic acid decorated polymers [92].

In a study by Zhang et al., it has been suggested that polyethylene glycol-grafted chitosan (PEG-g-chitosan) nanoparticles can improve insulin absorption at the systemic level following nasal administration [93]. As a crosslinking agent, tripolyphosphate ions were used to prepare insulin-loaded PEG-g-chitosan nanoparticles. In addition to being small (150–300 nm), the nanoparticles had a positive charge (+16 to +30 mV) and a high loading efficiency (20–39%). Release studies *in vitro* have shown that insulin was released in a burst, followed by a slow release over a prolonged period of time. When insulin-PEG-g-chitosan suspensions were administered intranasally to rabbits, their absorption of insulin was significantly improved compared with the absorption of insulin by the nasal mucosa of a suspension of insulin-PEG-g-chitosan alone and the control insulin solution without nanoparticles. With PEG-g-chitosan nanoparticles, it appears that the nasal mucosa can transport insulin [93].

#### 2.1.4. Pulmonary

Research on non-invasive drug delivery approaches, including pulmonary delivery systems, has been extended over the past few decades [94]. There are many benefits of using the lungs as a drug delivery site. In addition to a large absorptive surface area (80–120 m^2^), thin alveolar epithelium, low enzyme activity, proper vasculature and a short air-blood distance, lungs exhibit good vasculature [95,96]. The lung is relatively impervious to macromolecular drugs, including proteins and peptides, thereby posing a major obstacle to their widespread use. Many barriers inhibit the absorption of drugs from the lung into the bloodstream, including alveolar lining fluid layers, macrophages, and alveolar epithelial cells. Several potent and novel absorption enhancers have been used to enhance peptides and protein absorption in the lungs. In addition to absorption enhancers, five categories of enhancers have been identified, including protease inhibitors, liposomes, phospholipids, and cyclodextrins. It is possible, however, for these absorption enhancers to irreversibly damage alveolar epithelial cells. For the pulmonary delivery of macromolecular drugs, including protein and peptide drugs, a suitable absorption enhancer should be developed with high efficacy and low toxicity [97].

Poly-amidoamine dendrimer (PAMAM) is a synthetic polymer with a well-defined spherical shape. Dendrimer molecular weight and functional surface groups increase with dendrimer generation. They are known for their hydrophilic and highly water-soluble functional surface groups. Various drugs and macromolecules can be encapsulated and bound ionically or covalently to dendrimers by this structural feature. Drug delivery targets have used dendrimers as solubilizers and absorption enhancers. Increasing absorption through intravenous, oral, transdermal, and ocular administration is one of the main benefits of PAMAM dendrimers [98,99]. PAMAM dendrimers have been found to enhance the pulmonary absorption of proteins and peptides, but few studies have been performed on this relationship. The study by Dong et al. used insulin as a model of poorly absorbable proteins and peptides. They investigated different generations of PAMAM dendrimers for insulin absorption in the lungs. According to their findings, PAMAM dendrimers, especially G3 dendrimers, increased insulin absorption without damaging the membrane. In addition, various concentrations of G2 and G3 dendrimers enhanced pulmonary insulin absorption, as witnessed by the reduced plasma glucose level and plasma increase in insulin concentrations. All concentrations of G0 and 0.1% (*w*/*v*) of G1 dendrimer did not improve insulin absorption, but high concentrations of G1 dendrimer co-administered with insulin had a slight hypoglycemic effect. Based on the generation and concentration of insulin, dendrimers facilitate pulmonary absorption. Moreover, they may inhibit drug degradation as well as slow drug clearance from the lungs. Hence, they enhanced the pulmonary absorption of poorly absorbable proteins and peptides. The speculative mechanism was not supported by direct evidence, even though it has already been proven that could significantly reduce mucociliary clearance in nasal delivery systems. PAMAM dendrimers will require further research to reveal their absorption-enhancing mechanisms [97].

#### 2.1.5. Transdermal Delivery

The transdermal drug delivery (TDD) method is a highly effective method for releasing drugs into the systemic circulation. Transdermal absorption is easy to achieve due to the large surface area of the skin. Additionally, this technique is suitable for unconscious patients, patients who vomit, and patients who self-administer medication [100]. Compared to oral delivery, transdermal delivery has several advantages. TDD is safer than hypodermic injections, which may result in needle re-use and generate hazardous medical waste. As well as being non-invasive, these systems can be self-administered. It could provide a release for up to one week. Additionally, the systems are generally cheap and improve patient compliance [101]. It has been reported that microneedles containing biodegradable or dissolving polymers can be used for the transdermal delivery of protein drugs. Polymer microneedles can be made cheaper than silicon microneedles, and they do not pose a safety risk if they break off in the skin. This allows the needles to dissolve and degrade within the body safely. Compared to coated microneedles made of metal or silicon, polymer microneedles have a greater drug-loading capacity [102]. TDD is most effective for combination/concomitant diseases, such as diabetes and hypertension, because it avoids pre-systemic metabolism [103].

The composition characteristics of mesoporous silica nanoparticles (MSNs), such as high loading capacity, large specific surface area, facile surface functionalization, and high biocompatibility, have been extensively studied as therapeutic drug nanocarriers. As a result of physiological conditions, MSNs are highly stable [104,105,106,107]. Hollow mesoporous silica nanoparticles (HMSNs) have received more attention since they have large hollow interiors that could store more cargo than conventional silica nanoparticles [104,105,106]. In the work of Wang and his colleagues, HMSNs grafted with glucose-responsive polymer were used as a closed-loop glucose-sensitive antidiabetic delivery system. Additionally, the nanocarriers were integrated with poly(vinylpyrrolidone) nanocarriers for transdermal drug delivery. Drug storage was handled by HMSNs, and gatekeeping was provided by glucose-sensitive poly (3-acrylamidophenylboronic acid) (PAPBA) polymers. HMSN channels were effectively blocked by collapsed PAPBA chains after insertion into the skin at normoglycemic levels. On the other hand, hyperglycemic conditions resulted in an extended conformation and the emergence of nanopores, allowing the drug to be released. The glucose-sensitive antidiabetic delivery device could be controlled or programmed based on the physiological changes of diabetics since the nanopores of HMSNs-PAPBA are reversible. An effective transdermal delivery system for diabetic rats’ treatment with the designed glucose-responsive drug delivery system had a hypoglycemic effect comparable to that of subcutaneous injection. With this formulation, drug diffusion was effectively inhibited under normal blood glucose conditions, and the loaded drug was released in normal hyperglycemia conditions [108].

## 3. Polymeric Nanostructures for Improving the Delivery of Antidiabetics in Type 2 Diabetes (T2D)

Several studies have demonstrated that polymeric nanostructures can improve the efficacy of antidiabetic drugs in T2D by increasing the rate at which they are delivered. The nanostructures are made of biocompatible polymers, which allow them to contain antidiabetic drugs and release them in a controlled manner, resulting in improved drug delivery and better treatment outcomes. Several studies have proven the efficacy of polymeric nanostructures in improving the delivery of antidiabetic drugs for T2D patients [109,110]. There are several antidiabetic medications currently available for T2D, including metformin, sulfonylureas, glinides, DPP-4 inhibitors (DDP-4i), Glucagon-like peptide-1 (GLP-1) receptor agonists (GLP-1Ras), SGLT2 inhibitors (SGLT2i), and insulin. Polymeric nanostructures can improve their delivery. Indeed, although transient, some of the oral drugs provide some side effects to patients, mostly some gastrointestinal side effects, including nausea, vomiting, dyspepsia, abdominal cramps, diarrhea, and constipation. For example, diarrhea and nausea have been reported to occur in up to 30% of patients treated with metformin. Furthermore, diarrhea and nausea can occur in patients that take sulfonylurea, glinides and DDP-4i. The pain and discomfort lead the patients to stop the treatment and the reduction of therapeutic alternatives. In this regard, polymeric nanoparticles might offer some key advantages for improving the delivery of drugs while reducing the side effects. In addition, these nanostructures can increase drug uptake and distribution within the target tissue by encapsulating them and protecting them from degradation. The design of polymeric nanostructures can be tailored to target tissues, such as the liver or adipose tissues, where T2D leads to metabolic dysfunction. The aim of the ongoing research is to optimize nanoparticles for specific drug delivery applications and to test their safety and efficacy in human trials [109,111]. Here, we discuss the types of antidiabetic drugs and examples of polymeric nanostructures that could improve their delivery. Table 2 summarizes these drugs and the use of nanostructures to improve their drug delivery.

### 3.1. Biguanides

Treatment of T2D consists of lifestyle changes (weight loss, diet, and exercise) with the prescription of metformin, an antidiabetic class of biguanides that is the main first-line oral drug of choice in the management of diabetes [117,122,123]. Metformin is an insulin sensitizer drug that is currently the most prescribed antidiabetic medicine worldwide for T2D [124]. Several polymeric nanoparticles have been suggested as possible carriers for improving the delivery and efficiency of antidiabetic drugs. Several sugar-derived nanoparticles, such as alginate and chitosan, have either been used as encapsulating agents for metformin or as conjugates with the medicine to improve its bioavailability orally. Chitosan nanoparticles are a good candidate for integrating into metformin formulations to reduce the dose while maintaining the compound’s activity in preclinical studies [109,113,125]. A study by Mokhtare et al. used alginate (AL) and alginate-chitosan (AL-CS) beads to administer metformin [113]. Based on the results of this study, the amount and content of the drug contained in the beads were not significantly affected by the presence of chitosan in the bead formulation. It was also observed that pH affects drug release from AL beads, and long-term hypoglycemic effects were observed when AL- and AL-CS beads were administered orally to Diabetic Sprague Dawley rats. It has been shown that the pure drug decreases fasting blood glucose level for 3 h and then restores it after 3 h. Meanwhile, the fasting blood glucose level of metformin HCl-loaded AL-CS and AL beads peaked after 6 and 4 h, respectively. Therefore, the hypoglycaemic effect of both bead formulations was extended. AL and AL-CS beads containing metformin-HCl have been suggested to prolong the effects of metformin, thereby increasing patient compliance [113].

### 3.2. Sulfonylureas and Glinides

Other oral insulin sensitizers are thiazolidinediones (TZD), also called glitazones. The antidiabetic effects of TZD are mediated by PPAR nuclear receptors. TZD are efficient insulin sensitizers although they have been banned from various countries. When glycemic control is not well-achieved with metformin as monotherapy, then the insulin sensitizer can be accompanied by other oral antidiabetics as bitherapy. Among them are sulfonylureas (Sus) [126] and glinides, two popular classes of oral antidiabetics, which directly enhance insulin secretion in β-cells. Besides these insulin secretagogue drugs, there are oral inhibitors of glucose absorption and reabsorption. Inhibition of intestinal glucose absorption by alpha-glucosidase inhibitors such as acarbose and miglitol is one of the key strategies for lowering hyperglycemia in T2D [127].

O-carboxymethyl chitosan (O-CMC) nanoparticles (O-CMC NPs) are recognized for their effective biodegradability, biocompatibility, permeability, adhesion, solubilizing, and controlled release pattern [128,129]. In addition, O-CMC NPs exhibit good aqueous solubility, small particle size, and polymer swelling ability and are synergistic with chitosan antidiabetic effects. The efficiency of O-CMC NPs for improving antidiabetic delivery has been shown with glipizide (GPZ), a second-generation sulfonylurea drug that displays a low solubility and a short half-life (2–4 h). It has been found that GPZ coupled with O-CMC NPs (GPZ-O-CMC NPs) had a prolonged antidiabetic effect and higher release characteristics than pure and marketed GPZ. GPZ-O-CMC NPs significantly increased serum glucose, insulin, lipid profile (3–4 folds), oxidative stress markers (2–3 folds), and inflammatory cytokines (2.5–3.5 folds) in rats. Based on *in vitro* release studies, GPZ-O-CMC NPs released 22.5% of GPZ within two hours and 62.5% within 34 h. *In vivo* studies revealed that GPZ-O-CMCNPs significantly improved T2D-related biomarkers in comparison with pure and marketed GPZ, confirming their potential as a T2D treatment [117].

In a study published by Emami et al., glipizide-controlled release nanoparticles were synthesized using alginate and chitosan using an ionotropic-controlled gelation procedure. As a controlled delivery method for glipizide, alginate-chitosan nanoparticles (ACNP) are a convenient choice due to both the release-limiting properties of the system and the bioadhesive nature of the polymers. According to the results, the ionotropic controlled gelation method can be utilized in a mild aqueous environment to prepare glipizide-loaded alginate-chitosan nanoparticles (GlACNP) and can result in particles of favorable size, controlled release characteristics, and high entrapment efficiency that can be applied for glipizide delivery. Compared to currently available glipizide delivery systems, this novel formulation is expected to be a superior therapeutic alternative [115].

### 3.3. GLP-1 Receptor Agonists (GLP-1Ras)

GLP-1 receptor agonists (GLP-1Ras) are supposed to mimic GLP-1, the intestine hormone that enhances insulin secretion in response to glucose [130,131,132]. GLP-1Ras improve glucose-induced insulin secretion in similar mechanisms as GLP-1 does [133,134]. GLP-1Ras are issued from the modification of either native GLP-1 amino acid sequence as exemplified by Liraglutide or exendin 4, a natural GLP-1 mimetic extracted from the venom of the Gila monster lizard (*Heloderma suspectum*) [109,135]. Unlike all antidiabetics described above, GLP-1Ras are mainly delivered by subcutaneous injection with prefilled pens. The most adverse side effects associated with the use of GLP-1Ras are gastrointestinal symptoms, mainly nausea, along with injection site reactions. Therefore, some efforts are currently being made to switch the invasive mode of delivery of GLP-1RA to an oral route. These efforts are encouraged by the successful release of semaglutide, the first oral GLP-1RA dedicated to treat obesity [136].

Cationic polymers, such as linear polyethylenimine (lPEI), an excellent vector for nucleic acid delivery [137], have been considered for GLP-1 therapy in T2D [120]. In the work of Nie et al., the efficiency of lPEI-based NPs has been demonstrated for delivering a plasmid DNA encoding GLP1 [120]. To do so, the lPEI has been made hydrophilic and electrostatically neutral by coating the nanoparticle surface with 1,2-dipalmitoyl-sn-glycerol-3-phosphocholine (DPPC) and 1,2-dimyristoyl-rac-glycero-3-methoxy poly (ethylene glycol)-2000 (DMG-PEG). Thanks to this modification, the lPEI/DNA complex had improved diffusivity and transport properties in mucus. In addition, a single dose of these modified nanoparticles increased GLP-1 expression for over 24 h in the liver, lungs, and intestine of a mouse model of T2D. Injection of the single dose improved blood glucose levels within normal ranges for at least 18 h [120].

### 3.4. Dipeptidyl Peptidase-4 (DDP-4) Inhibitors

Other oral antidiabetics are the dipeptidyl peptidase-4 (DPP-4) inhibitors (DPP-4i), with sitagliptin, saxagliptin, and linagliptin as leading compounds in the market. DPP-4i inhibits DPP-4, which is the enzyme that digests glucagon-like peptide-1 (GLP-1) [138]. Inhibition of DPP-4 is expected to increase GLP-1 activity, which in turn stimulates insulin secretion. A composite system for dual-drug delivery was described by Arajo et al., in which GLP-1 and a DPP-4 inhibitor (iDPP4) are combined [121]. A droplet microfluidics technique was used to assemble this system, which involves the use of polymeric poly (lactic-co-glycolic acid) (PLGA) nanoparticles. In addition to providing a protective, stable, and biocompatible environment for the encapsulated peptides and proteins, these nanoparticles were functionalized with chitosan (CS) and a cell-penetrating peptide (CPP). The study used a rat model of T2D induced by streptozotocin and nicotinamide, a non-obese model of T2DM. As a result of combining both drugs, GLP-1’s hypoglycemic effects were significantly increased, but they were sustained and prolonged, while iDPP4 improved the drug’s therapeutic efficacy. Four hours after oral administration of the system, blood glucose levels dropped by 44% and remained constant for another four hours. The dual-drug composite system also enhanced plasmatic insulin levels 6 h after oral administration, and pancreatic insulin levels were also higher. This is a promising result for the oral delivery of GLP-1, which should be investigated further in chronic diabetic models [121].

### 3.5. Sodium—Glucose Co-Transporter 2 Inhibitors (SGLT2i)

Blocking the renal reabsorption of glucose is the other strategy to combat hyperglycemia. This is achieved by inhibitors of sodium-glucose co-transporter 2 (SGLT2i). Represented by several members, including canagliflozin, dapagliflozin, empagliflozin, and meglitinide, SGLT2i also promote urinary glucose excretion and protects patients from cardiovascular risks. Even though SGLT2i has made substantial progress since its discovery, they remain poorly water-soluble. By improving aqueous solubility with nanoparticles, SGLT2i patients with T2D might be able to reduce the dose of oral delivery, experience fewer adverse effects, and also have more bioavailability [109]. As a result, there have been very few studies that have been published to date. The use of lipids and surfactants in a nanoparticle-based formulation of dapagliflozin has been investigated in one study. It has been shown that dapagliflozin has the best *in vitro* permeability and sustained release when mixed with tween 20 and labrasol, respectively. This study may pave the way for further preclinical studies [109,139]. Future drug delivery of this kind of drug could be improved by combining it with polymer nanostructures.

## 4. Biomedical Imaging

A total of three distinct areas of biomedical imaging have evolved rapidly: imaging molecular biomarkers, single cells, and therapeutics. Each of these areas has great potential for accelerating progress [140]. The development of strategies for optimal diabetes prevention and treatment will depend on clinical imaging of β-cell mass (BCM). Non-invasive monitoring of BCM and its function has attracted considerable interest in recent years [141]. As mentioned in the previous section, it has been demonstrated that both insufficient numbers and/or functional decline of β-cells are critical components in the development and progression of diabetes and hyperglycemia. For clinical purposes, early detection of β-cell changes is crucial for the timely diagnosis and treatment of diabetes, especially during the initial stages of the β-cell compensation phase (before glucose levels become elevated). Hence, new tools that could non-invasively map BCM are urgently needed. The most promising technique is cell imaging [142,143,144]. Researchers have investigated several potential targeted β-cells biomarkers, including glucagon-like peptide 1, vesicular monoamine transporter 2, presynaptic vesicular acetylcholine transporter, and sulfonylurea receptor [145,146,147,148].

The tiny Langerhans cells contain β-cells, which make up 1–3% of the pancreatic mass. Because of their dispersed localization (unlike tumor cells), they are difficult to visualize. Furthermore, in order to determine differences in anatomical BCM between diabetic subjects, all intact islets remaining in the later stages of diabetes must be detectable. Therefore, detection methods should be highly sensitive. The following section will examine fluorescent imaging and magnetic resonance imaging (MRI) as methods for detecting β-cells. In Table 3, the characteristics of these two methods are briefly compared.

### 4.1. Fluorescence Imaging

The use of optical imaging is less expensive and more convenient for every application as long as the target tissues are in a convenient location. The software used to operate optical imaging equipment and analyze optical imaging data is easy to use. Typically, researchers can perform both bioluminescence and fluorescence imaging without the assistance of staff with special expertise. There are many advantages to fluorescence imaging, including high-throughput screening for compound optimization, high sensitivity, and multi-color imaging, as well as its shortcomings, including lack of clinical translation and low-depth penetration [144,149,150]. Using fluorescence imaging techniques with polymeric nanoparticles, researchers could monitor drug pharmacokinetics and drug delivery [151]. By using the photons emitted from fluorescent probes, fluorescence imaging allows the non-invasive visualization of cells, molecules, tissues, and organs of a body [152].

Sulfonylurea receptor 1 (SUR1) is a class of compounds that are expressed on the surface of β-cells and other islets, similar to glucagon-like peptide-1 (GLP-1) receptor agonists, which is largely expressed in the β-cells’ secretory granules. A sulfonylurea, known as glibenclamide, has been used since the 1960s as a β-cells imaging ligand, owing to its affinity for SUR1, which made it suitable for T2D patients [153,154,155]. By using multivalent derivatives of the antidiabetic drug glibenclamide, Babi et al. were able to mimic β-cells by targeting the SUR1. Since glibenclamide binds strongly to SUR1 but does not possess a functional group suitable for linking to an imaging probe, 11 derivatives of glibenclamide were synthesized and evaluated in MIN6 cells for their affinity for SUR1. It has been reported that the most promising compound in the multivalent glibenclamide-polyamidoamine (PAMAM) family contains 15 sulfonylureas per dendrimer. The researchers synthesized fluorochrome-labeled multivalent probes to demonstrate better binding affinity to SUR1 than native glibenclamide in cellular assays. The persistent release of insulin from MIN6 cells exposed to high glucose concentrations proved that the multivalent probes had very low cytotoxicity. β-cells from primary mouse and human islets, as well as Langerhans islets, were also positively labeled by the probes. The probes were synthesized by attaching rhodamine-X-NHS to the PAMAM-G5 backbone to facilitate fluorescence imaging. A reliable and repeatable conjugation of the glibenclamide derivative 12 to PAMAM dendrimers was accomplished after activation with HATU/DIPEA. As a result, the ligand loading was well-controlled. For the establishment of the correlation between avidity and multivalency, NMR and MALDI-MS were used to characterize final probes for rhodamine X and the targeting moiety. An overview of this strategy is depicted in Figure 8 [156].

### 4.2. Magnetic Resonance Imaging (MRI)

MRI is a non-ionization imaging technique in which water nuclei (primarily protons) are measured by their relaxation rates either longitudinally or transversely (1/T1 or 1/T2) [157]. With contrast agents, it is possible to provide high-resolution, three-dimensional images of soft tissues by enhancing the contrast between normal and diseased tissues. In MRI, paramagnetic metal chelates, such as Gd (III), Fe (III), and Mn (II) complexes, and ultrasmall super magnetic iron oxide enhance image contrast by altering water protons’ relaxation rates [158,159,160]. The high soft tissue contrast of MRI makes it ideal for *in vivo* imaging of pancreatic islets, as well as single-photon emission computed tomography (SPECT) and positron emission tomography (PET). Under clinical conditions, computed tomography and MRI cannot resolve single islets [161,162,163,164,165]. A contrast agent can be used to enhance the contrast between the exocrine and endocrine pancreas using PET and SPECT imaging or MRI to detect β-cell-specific metabolites; however, it is much less sensitive [161,162,163,166].

In MRI and computed tomography, less than 100 μm of spatial resolution could be used to delineate the pancreas [167]. This method, despite the fact that the pancreas is an extremely vascularized soft organ and the islets of Langerhans only constitute 2 to 3% of the pancreas’ tissues, makes it virtually impossible to distinguish the islets from other pancreatic tissues due to their high vascularization level. A key to visualizing β-cells in the islets of Langerhans is developing agents that recognize and respond to β-cells’ biological functions [168,169,170].

A number of MRI contrast agents are capable of being carried by polymeric nanoparticles. MRI is an efficient modality for tracking lymphocytes *in vivo*. Labeling strategies could be antigen-unspecific or antigen-specific. The immune system invades the islets during the initiation phase of T1D, causing β-cells death. As a consequence, non-invasive *in vivo* visualization of infiltrating lymphocytes may allow early detection of pre-diabetic patients [171]. Contrast agents accumulating in islets or being chemically altered as a result of metabolic activity have been shown to yield promising results *in vivo* MRI. As a result of the high concentrations of these compounds required for MRI, human studies remain challenging due to potential toxic effects [172,173]. While MRI offers a number of benefits, such as clinical translation, high resolution, and soft-tissue contrast, it also has several disadvantages, such as high cost and lengthy imaging times [174,175,176].

Recently, new methods have been developed to assess pancreatic β-cells that have overcome MRI’s limited sensitivity. By enhancing sensitivity by Moore and coworkers’ method in 2002, they distinguished endocrine from exocrine cells and monitored pancreatic islet cell infiltration for the first time [171]. BCM and function could be analyzed *in vivo* with Mn^2+^-enhanced MRI. Only functional cells with efficient glucose metabolism will take up high levels of Mn^2+^ since uptake is glucose-dependent [173]. As glucose enters the β-cell, it is metabolized, which increases the intracellular concentration of adenine nucleotides (ATP) and inhibits K^+^ channels. It is voltage-gated L-type Ca^2+^ channels that regulate insulin secretion. Mn^2+^ is an excellent contrast agent for NMR due to its T1 relaxation properties. It was first used for MRI contrast in addition to assessing the potency of isolated islets *in vitro* [177]. It is also suitable for evaluating the function of endogenous and grafted pancreatic islets *in vivo*, as demonstrated in Figure 9 [161,178,179]. With Mn^2+^ as a paramagnetic surrogate of Ca^2+^, Roman and coworkers developed an enhanced MRI method to assess β-cell function in cultured or isolated mouse islets. In T1-weighted MRI images, exogenous Mn^2+^ takes up Ca^2+^ channels after β-cells or islets are stimulated with D-glucose for insulin secretion [180]. Islet transplants can also be labeled and longitudinally monitored non-invasively using superparamagnetic iron oxide nanoparticles (SPION). A study by Saudek and coworkers used commercial SPION to label β-cell islets and transplant them into rats’ livers via the portal vein [181,182].

The use of SPIONs or ultra-small superparamagnetic iron oxide nanoparticles (USPIONs) in bioscience and clinical research has increased dramatically in recent decades, including imaging, targeted drugs, and gene delivery [183]. According to Yang et al., a non-toxic and simple synthesis approach was developed for the preparation of USPIONs coated with an amphiphilic polymer (carboxylated polyethylene glycol monooleyl ether [OE-PEGCOOH]), a proprietary method for producing water-soluble USPIONs using hydrated ferric salts as the iron precursor (instead of toxic ferric acrylacetone [Fe(acac)_3_]). Next, these nanoparticles were functionalized with monoclonal antibodies (β-cell lymphoma). Using these nanoparticles, β-cells could internalize and label primary islet cells at relatively low iron concentrations. Comparing these products with FeraSpinTM S, a commercial product from USPION, they were found to be biocompatible and non-cytotoxic. MRI scanners in clinical settings with a field strength of 3.0 Tesla were able to detect MRI *in vivo*, despite some cases showing hypo intensity changes at the transplant site. The signal intensity of the tracer decreased gradually from day one to day 21 during the immune rejection period. Fe concentration variations also affected signal intensity. The excessive dose of 30 g/mL used for labeling the 150 rodent islet cells resulted in a hypointense signal at first. Long-term monitoring was not possible because signals disappeared at 10 g/mL within a short amount of time. The signal elevation around the transplantation site was caused by inflammatory liquid infusions, which frequently occur during transplant surgery. In addition to Prussian blue staining and immunohistochemistry staining, iron deposition and attachment in islets could be confirmed [184].

## 5. Wound Dressing

A diabetic wound is a severe injury that frequently occurs in diabetic patients. In diabetes, foot and leg ulcers can lead to hospitalizations and limb amputations, as wound healing times are long. Diabetes-related limb amputations account for 50–70% of all amputations [185,186]. A number of interesting properties of polymer wound dressings make them useful for the treatment of chronic wounds, especially diabetic wounds. The ideal polymeric dressing should have a high porosity and swelling capacity, as well as adequate water vapor transmission rates (WVTR), moisture, warmth, gaseous permeation, excellent antimicrobial properties, superior mechanical performance, and bioactive agents [187,188]. Diabetic wounds could also be poorly healed with crosslinked dressings because of their poor wound-healing properties and biological activity. The encapsulation of bioactive agents in polymer-based dressings has been shown to be a promising approach to wound care, particularly chronic wounds. Bioactive agents in wound healing applications include antibiotics, growth factors, stem cells, plant extracts, antioxidants, anti-inflammatory drugs (e.g., curcumin, etc.), and vitamins. Transdermal patches, hydrogels, foams, membranes, films, and nanofibers are polymeric wound dressings that can contain bioactive agents [189].

### 5.1. Classification of Wound Dressings

There are a number of ways wound dressings can be used in the treatment of various injuries. Their primary function is to protect wounds from bacteria and to speed up wound healing. The majority of wound dressings available today fail to provide moisture, delay healing, and cause allergic reactions. Thus, effective wound dressings are urgently needed. The four major types of wound dressings are traditional/passive, interactive, skin substitutes, and bioactive. A traditional dressing protects an injury from foreign objects and contamination, stops bleeding, cushions the wound, and absorbs wound exudate. When removed, these dressings can cause bacterial infections and harm to the skin. Interactive dressings like composites, films, gels, foams, and sprays promote wound healing by providing a moist environment, exhibiting efficient water transmission, and enhancing re-epithelialization and granulation. Bioactive agents can also be added. It is possible to regenerate skin with tissue-engineered scaffolds, such as Apligraf, OrCel, and TransCyte, derived from co-cultured cells or cell-seeded scaffold material. As a result of the possibility of wound infections, disease transmission, rejection by the body, cost, and limited shelf-life of dermal grafts, they are one of the most essential materials in dermatology and plastic surgery. Allografts, autografts, and xenografts are examples of dermal grafts. These materials can also be used for traumatic wounds, burn reconstruction, oncologic defects, vitiligo, scar contracture release, and hair restoration. The problem is that they cannot handle complex injuries (those with exposed bones and deep spaces). Bioactive dressings include hydrocolloids, sponges, wafers, foams, nanofibers, hydrogels, collagen, and films, and they are biodegradable, biocompatible, and can be used as drug delivery systems. They include nanoparticles, GFs, vitamins, and antibiotics. The use of nanofibers and hydrogels as wound dressings can be enhanced by delivering drugs in controlled ways [43,190,191,192].

### 5.2. Polymer-Based Dressings for Diabetic Wound Management

A cerium oxide nanoparticle-chitosan/cellulose acetate film has been reported by Kalaycıoğlu et al. [187]. According to this study, the films exhibited antibacterial properties, improved mechanical and thermal features, and appropriate UV permeability inhibition, WVTR, and pH properties for wound dressings. With the combination of chitosan and cellulose acetate, a hydrophilic, cell-attachable, and proliferation-promoting composite was formed. A transition from Ce^3+^ to Ce^4+^ in cerium oxide nanoparticles is being considered for biomedical applications. The addition of cerium oxide nanoparticles improved antibacterial, thermal, and mechanical properties as well as antioxidant, antibacterial, anti-inflammatory, and biological mimicry properties that helped to accelerate the healing of diabetic wounds. Films were produced in one step without complicated synthesis techniques, making them a potential wound dressing. Using the solvent-casting method, the composite was formed by using chitosan and cellulose acetate, both of which dissolve in formic acid [187]. Nanofiber mats containing chitosan, polyvinyl alcohol (PVA), or zinc oxide (ZnO) are beneficial due to their therapeutic properties, which make them useful in dressings for diabetic wounds [193]. In the work of Ahmed et al., chitosan, ZnO nanoparticles, and PVA were used in diabetic wound healing. Chitosan releases N-acetyl D-glucosamine, which promotes fibroblast proliferation, collagen synthesis, and the deposition of hyaluronic acid. Furthermore, it could stimulate cell adhesion, proliferation, or wound healing as it mimics the extracellular matrix (ECM). ZnO nanoparticles are used as an active ingredient in wound dressings, due to their antimicrobial properties and their role in fibroblast proliferation and angiogenesis [194]. As part of the composite nanofiber membrane formation process, PVA is also added to chitosan to help improve its mechanical, biodegradable, and hydrophilic properties. As a result of the presence of PVA in the chitosan nanofibers, cell viability, proliferation, and gene expression are enhanced, contributing to the biocompatibility of electrospun membranes. Compared to chitosan/PVA nanofibrous mats, chitosan/PVA/ZnO nanofibrous membranes had a higher antibacterial capacity against *E. coli*, *P. aeruginosa*, *B. subtilis*, and *S. aureus*. Compared to chitosan/PVA nanofibers, chitosan/PVA/ZnO nanofibrous membranes exhibited improved wound healing in T2D, which is a major problem for T2D patients worldwide. As illustrated in Figure 10, in the diabetic rabbits treated with chitosan/PVA/ZnO nanofibrous membrane, wound closure was significantly different on days 8 and 12. A 90% wound closure rate was observed in the chitosan/PVA/ZnO nanofibrous membrane-treated group on day 12, whereas only a 52.3% wound closure rate was achieved in the control group [195].

In the work of Tan et al., the aim was to evaluate the efficacy of a novel formulation of hydrocolloid dressing, a sodium alginate (SA) biomaterial scaffold with vicenin-2 (VCN-2), for the treatment of an excision wound in T1D. A variety of crosslinking degrees in sodium alginate makes it an effective drug delivery vehicle. By controlling the slow-release properties of bioactive materials (i.e., drugs), it reduces the swelling of hydrocolloid films when exposed to water (wound exudate) and ensures high formulation effectiveness [196]. Numerous studies have examined the pharmacological effects of flavonoid glycosides, including VCN-2. Diabetic wound healing may be facilitated by VCN-2 via VEGF and TGF-β signaling pathways. A combination of VEGF and TGF-c growth factors and fibroblast migration and infiltration contributes to wound healing [197,198]. The underlying mechanism by which VCN-2 promotes wound healing has been investigated *in vivo* using hyperglycemic rats. As a result, VCN-2 hydrocolloid films may have tremendous potential for enhancing diabetic skin wound healing in the future [197]. In the past decade, stem cell-based treatments for diabetic wounds have proven significant promise. Due to the hostile wound environment, stem cell therapy is severely limited in terms of clinical translation. Diabetic wound repair patients have been able to increase the engraftment of transplanted stem cells by using hydrogel systems [199,200].

Hyaluronic acid (HA) consists of D-glucuronic acid and (1-b-3) *N*-acetyl-D-glucosamine alternating units. Functional sites in the HA molecules allow crosslinking, ligand conjugation, and other configurations, making them useful for therapeutics. As well as being non-immunogenic, biodegradable, and viscoelastic, HA is ideal for nanomedicine, vaccines, mucosal delivery, surgical aids, wound repair, and regeneration. In normal tissues, HA reduces cytotoxicity and side effects. Extracellular matrix proteolytic enzymes (MMPs) and nitric oxide (NO) in the body degrade HA. As a targeting moiety, HA can be conjugated to nanoparticles, dendrimers, carbon nanotubes, graphene, and quantum dots [41,201,202,203,204].

Xu and his colleagues have developed a new generation of in situ-formed injectable hydrogel systems composed of thiolated hyaluronic acid (HA-SH) in combination with hyperbranched polyethylene glycol (PEG) macromers (HP-PEGs). To achieve an effective healing process, a combination of a synthetic polymer and a naturally derived polymer could be used to fabricate *in situ*-formed injectable hydrogel systems. Without the need for complex methods, injectable hydrogel could be applied to any irregular shape during hydrogel formation. Cell culture scaffolds and tissue engineering have extensively used PEG and its derivatives for the fabrication of biocompatible hydrogels. To produce HP-PEGs with pendent acrylate groups, a reversible addition-fragmentation chain transfer (RAFT) precursor agent, disulfiram (DS), was utilized. By crosslinking HA-SH with HP-PEGs, it was possible to control their mechanical properties, swelling and degradation profiles, and antifouling properties. For the purpose of testing HP-PEG/HA-SH/ADSCs for the healing of diabetic wounds, rats were used. As a result of the hydrogel system encapsulating ADSCs, diabetic wound healing processes were accelerated due to its ability to suppress inflammation, promote angiogenesis, and re-epithelialize. As depicted in Figure 11, HP-PEG hydrogels could be synthesized by combining 2,2′-azobis (2-methylpropionitrile) (AIBN) and DS as initiators for *in situ* RAFT polymerization. ADSCs are embedded in the hydrogel after HA-SH and HP-PEG are mixed *in situ*. Humanized diabetic wound models are used to test the hydrogel system. A hydrogel system encapsulated with ADSCs was capable of accelerating diabetic wound healing by inhibiting inflammation, increasing angiogenesis, and re-epithelializing diabetic wounds in a beneficial manner [205]. Table 4 summarizes some of the latest advances in wound dressings using polymer-based nanostructures.

## 6. Conclusions

Diabetes has become one of the top non-communicable health problems in the 21st century. Most developing and newly industrialized countries have epidemics of the disease, and it is the leading cause of premature death and disabilities in developed countries. Currently, nanotechnology combined with other technologies to manage diseases and diabetes is not an exception. There are several interesting properties of polymer nanocomposites that make them useful in diabetes management. Various natural and synthetic polymers are important for designing nanostructured delivery systems, improving imaging, and enhancing wound healing. In the past decade, polymer-based nanostructures have achieved rapid advances in imaging and engineering, pointing to a promising future. Thus, it makes sense that in the field of diabetes, more research needs to be conducted regarding the use of these materials, especially natural polymers, for measuring β-cell mass (BCM). The opportunity for further imaging is extraordinary. A number of advances have been achieved in nanoparticle formulations for nano-insulin delivery. Utilizing nano-polymeric systems, these advances have made nanoparticle formulations much more effective at delivering high concentrations of insulin in a highly bioavailable manner. Polymer-based nanostructures are capable of delivering insulin and antidiabetic drugs into the intracellular space. The high encapsulation efficiency and biocompatibility of polymer nanoparticles allow them to target intracellular compartments and provide sustained release kinetics, making them ideal carriers. However, these non-invasive routes of administration have limitations, including poor bioavailability, inadequate penetration through physical and chemical barriers, and open-loop administration that leads to poor blood glucose regulation. By combining polymer chemistry and nanotechnology with these drugs, therapeutic efficacy can be improved.

Diabetic wound management can be made easier with polymer-based wound dressings because of their several interesting properties. In addition to high porosity, high water uptake and swelling capacity, moderate water vapor transmission rates, acceptable mechanical performance, gaseous exchange, and fast wound healing, some of these properties are also beneficial. Despite their potential to manage diabetes, polymers also have downsides. Effective diabetes management requires biocompatible polymers. Some polymers trigger inflammation or immunity. Polymer matrix drug release rates can be difficult to control, resulting in over- or under-doses of medication. Polymers degrade over time, affecting drug release and efficacy. Temperature, pH, and humidity may also affect polymer-based formulations’ properties and stability over time.

The purpose of this review was to briefly discuss the use of polymer-based nanostructures in the management of diabetes that can facilitate treatment and diagnosis in three areas: imaging, drug delivery and wound healing. In addition to assessing the effects of polymer nanostructures on cell viability and function, nanostructures can also be analyzed histologically and biochemically. A lack of long-term studies on the safety and efficacy of polymer-based drug delivery systems makes it difficult to assess their long-term benefits and risks. Preclinical and clinical studies are necessary to determine polymer nanostructures’ safety and effectiveness. The pharmacokinetics, biodistribution, toxicity, and efficacy of delivery systems can be critical for improving patient outcomes. Even though polymers have shown promise for diabetes management, numerous challenges remain to be addressed to realize their full potential.

## Figures and Tables

**Figure 1 pharmaceutics-15-01215-f001:**
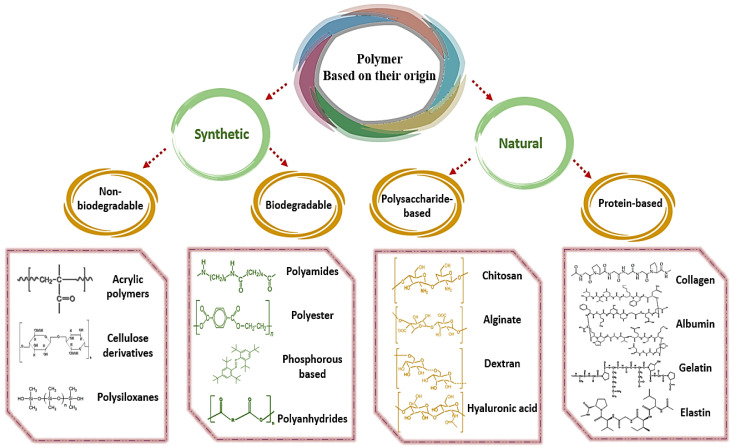
Classification of polymers-based on their origin.

**Figure 2 pharmaceutics-15-01215-f002:**
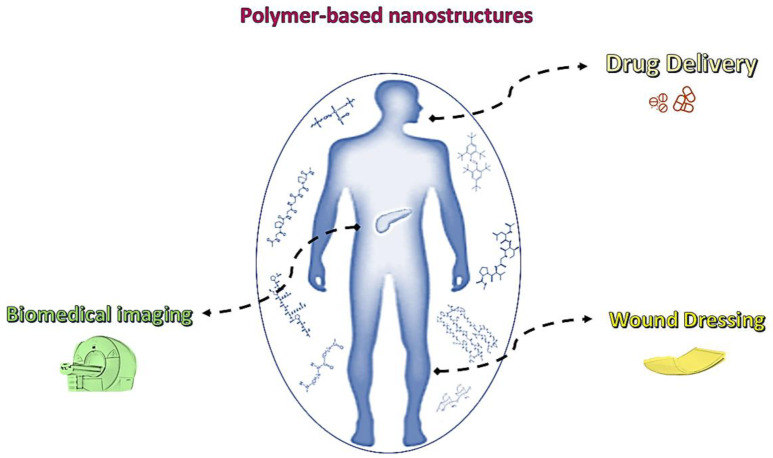
Overview of ways polymer-based nanostructures can benefit diabetes management. A variety of polymer-based nanostructures can be used to manage diabetes, including fluorescence and magnetic resonance imaging to detect β-cells; drug delivery via oral, buccal, nasal, pulmonary, and transdermal routes; wound healing.

**Figure 3 pharmaceutics-15-01215-f003:**
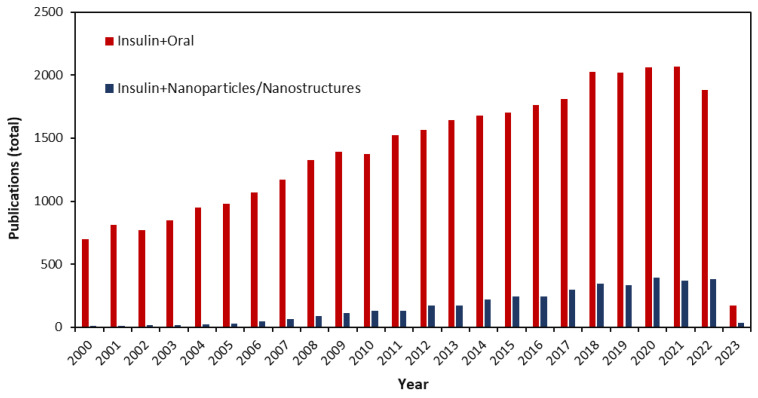
The number of articles published in the field of insulin drug delivery by oral administration and insulin nanoparticles/nanostructures between 2000 and 2023.

**Figure 4 pharmaceutics-15-01215-f004:**
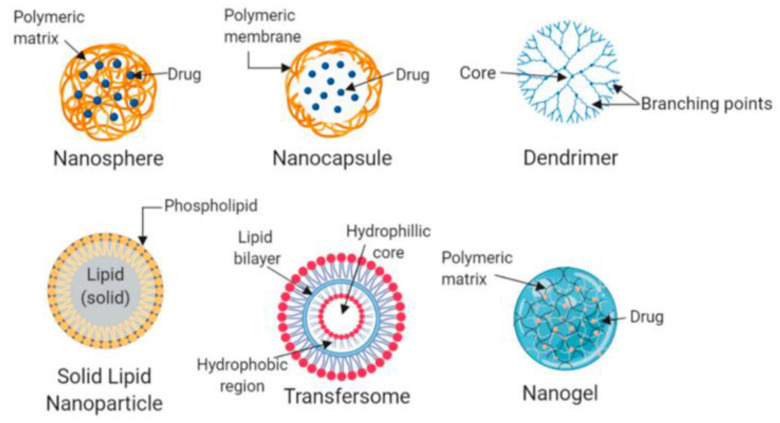
Various platforms for insulin loading and delivery have been explored in the literature, including nanospheres, nanocapsules, dendrimers, solid lipid nanoparticles, transfersomes, and nanogels [44].

**Figure 5 pharmaceutics-15-01215-f005:**
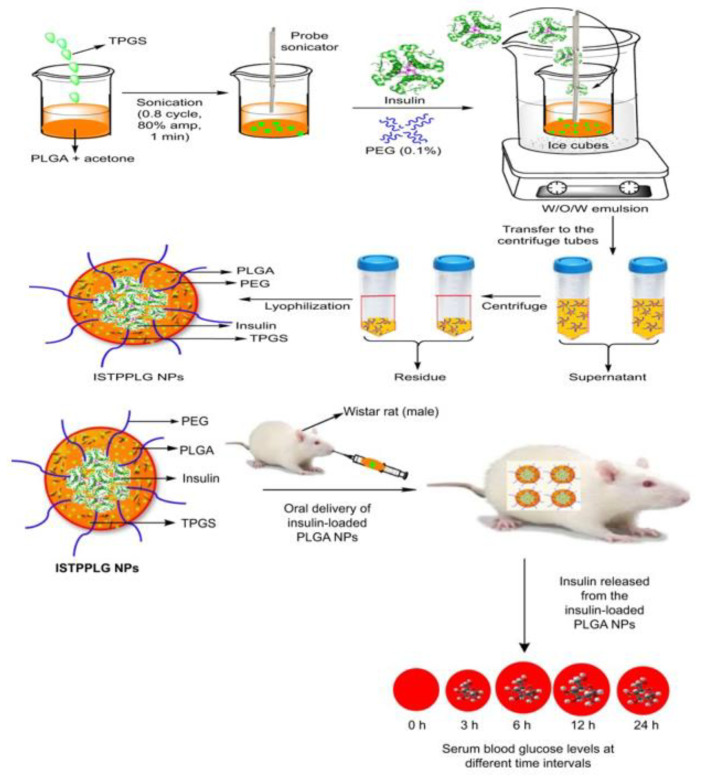
TPGS-emulsified PEG-capped PLGA nanoparticles loaded with insulin are synthesized using TPGS-emulsified PEG-capped PEG, and insulin release from PEG-capped TPGS-emulsified PLGA nanoparticles containing insulin-loaded TPGS-emulsified TPGS nanoparticles [79].

**Figure 6 pharmaceutics-15-01215-f006:**
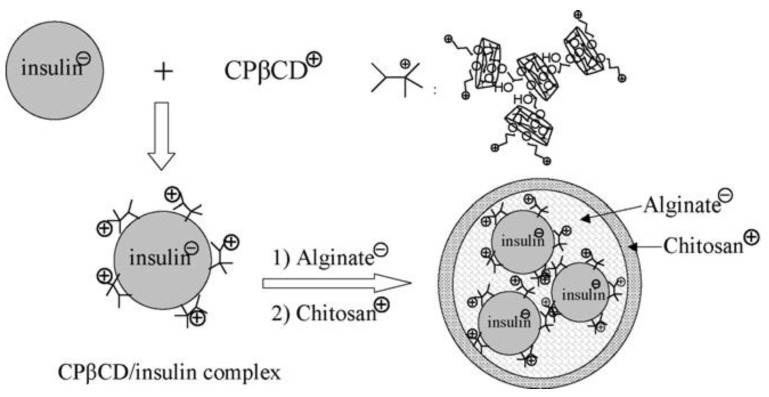
This schematic shows how alginate/chitosan nanoparticles interact with insulin and its protection from oxidation by CPβCDs [84].

**Figure 7 pharmaceutics-15-01215-f007:**
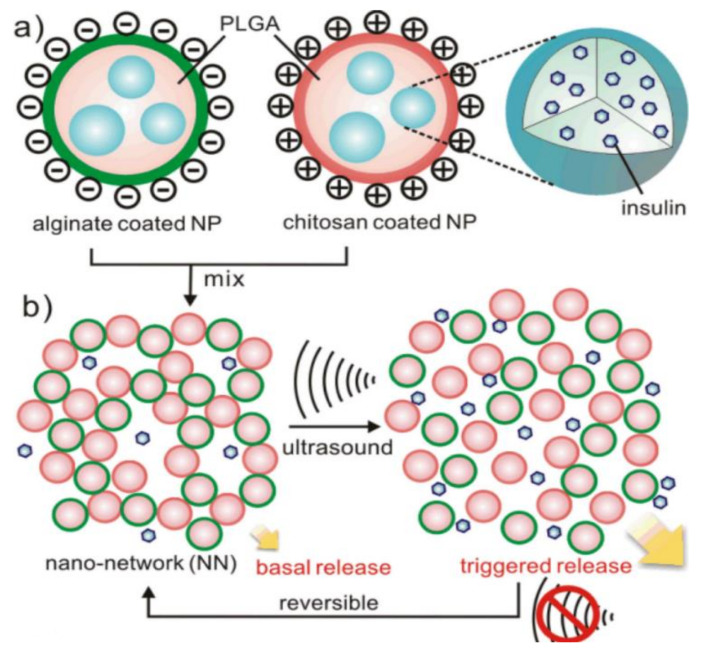
(**a**) Nanoparticles encapsulating insulin are embedded in PLGA and coated with alginate and chitosan. (**b**) A nano-network is formed by combining oppositely charged nanoparticles. It triggers the dissociation of the nano-network and promotes insulin release [85].

**Figure 8 pharmaceutics-15-01215-f008:**
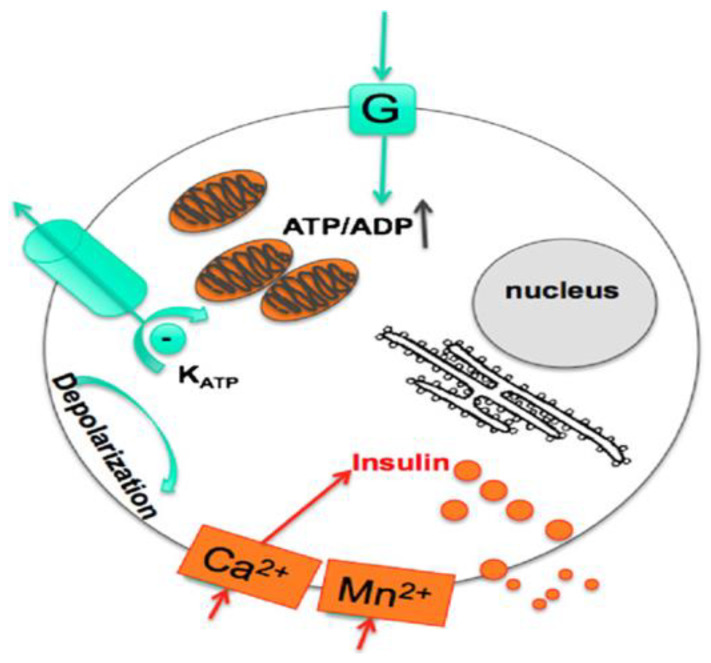
Glucose-dependent Ca^2+^ channels participate in the entry of Mn^2+^ into β-cells through these channels [156].

**Figure 9 pharmaceutics-15-01215-f009:**
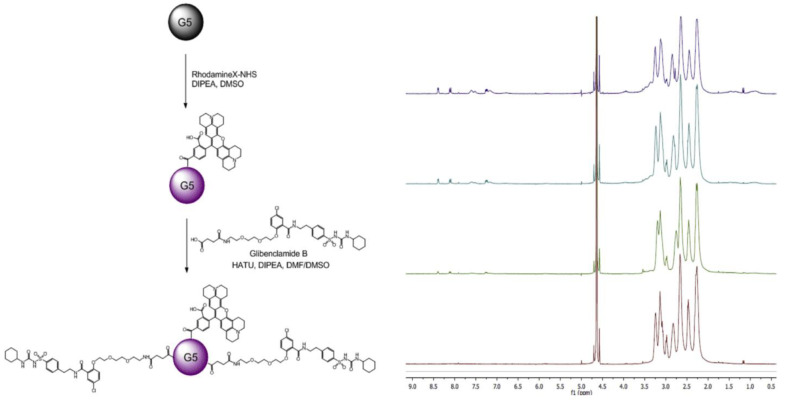
An overview of the synthesis of PAMAM-rhodamine-X-glibenclamide conjugates and a comparison of the ^1^H NMR spectra of conjugate probes containing increasing amounts of glibenclamide loading of 0, 2, 5, 15 glibenclamide ligands 12 per dendrimer from bottom to top (right) [161].

**Figure 10 pharmaceutics-15-01215-f010:**
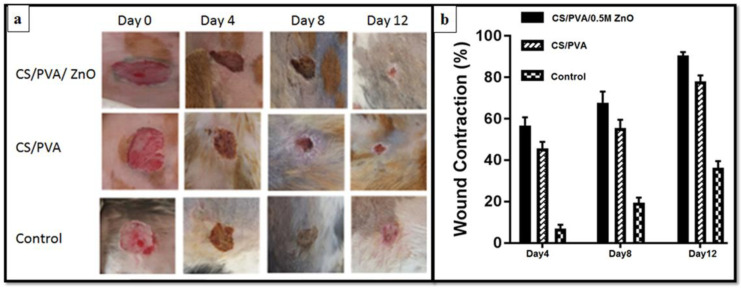
(**a**) The wound healing time of diabetic rabbits treated with control, CS/PVA, and CS/PVA/ZnO nanofiber mats over a 12-day period. (**b**) A chart illustrating 12 days of wound contraction [195].

**Figure 11 pharmaceutics-15-01215-f011:**
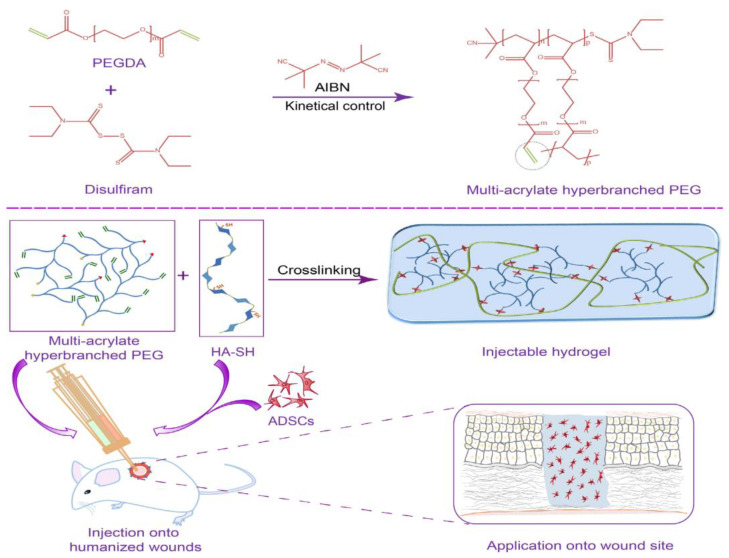
An overview of the concept of creating and utilizing injectable HP-PEG hydrogels. By combining AIBN and DS as initiators, *in situ* RAFT polymerization could be used to synthesize HP-PEG polymers. After HA-SH and HP-PEG are mixed *in situ*, ADSCs are embedded in the hydrogel. A humanized diabetic wound model is then used to test the hydrogel system [205].

**Table 1 pharmaceutics-15-01215-t001:** A brief summary of the various administration routes.

Delivery Approach	Positive Aspects	Negative Aspects	Barrier
Oral	No invasive procedure	Degradation in first-pass metabolism	Enzymes, acids, proteases, and cells of the epithelium
	Self-administration is possible		
Buccal	No invasive procedure	Absorption surface is low Limiting total dose	Cells of the epithelium and mucus
	Self-administration is possible		
	Absorptive speed		
Nasal	Non-invasive procedure	Absorption surface is low Limiting total dose	Cells of the epithelium and mucus
	Self-administration is possible		
	Absorptive speed		
Pulmonary	Non-invasive procedure	Patient training is required to prevent variability in dosing with inhalers	Cells of the epithelium and mucus, surfactant
	Absorptive speed		
	Absorption surface area is large		
Transdermal	Non-invasive procedure	Absorption is slow Skin permeability is low Problems related to infection	Shedding of cells on a constant basis and layers of the *stratum corneum*
	Self-administration is possible		
	Time-controlled release is possible		

**Table 2 pharmaceutics-15-01215-t002:** An overview of antidiabetic drugs, their mechanisms of action, and examples of nanostructures to improve their drug delivery.

Drug Class	Example of Drug	Mechanism	Polymer-Nanostructures	Ref
Biguanides	Metformin	Boosts insulin sensitivity and reduces glucose production by the liver	Alginate	[112]
			Alginate/chitosan complex	[113]
			Hyaluronic acid polymer	[114]
Sulfonylureas and Glinides	Glipizide, glyburide, glimepiride	Produces insulin by stimulating the pancreas	Alginate	[115]
			Poly lactic-co-glycolic acid	[116]
			O-carboxymethyl chitosan	[117]
GLP-1 Receptor Agonists	Exenatide, liraglutide, dulaglutide	Increases insulin production and decreases liver glucose production by activating GLP-1 receptors	Poly (DL-lactic-acid-co-glycolic acid) (PLGA)	[118]

			PLGA with chitosan	[119]
			linear polyethylenimine	[120]
Dipeptidyl Peptidase-4 Inhibitors	Sitagliptin, linagliptin, saxagliptin	Increases insulin production by inhibiting DPP-4 enzyme		
			Chitosan conjugated PLGA	[121]
Sodium—Glucose Co-transporter 2 Inhibitors	Canagliflozin, dapagliflozin, empagliflozin	It increases glucose excretion in the urine by blocking glucose reabsorption	-	

**Table 3 pharmaceutics-15-01215-t003:** A comparison of fluorescence and magnetic resonance imaging techniques.

	Fluorescence Imaging	Magnetic Resonance Imaging
Type of energy measured	Visible or near-infrared light	Radio waves
Depth	<5 mm	No limit
Sensitivity	nM	µM-mM
Spatial resolution	100 nm	50–100 µm
Type of contrast agent	Fluorescence molecules	Small molecules, nanoparticles
Advantage	High throughput	High soft tissue contrast
Drawback	No anatomical information	Low throughput

**Table 4 pharmaceutics-15-01215-t004:** A summary of recent advances in wound dressings based on polymer-based nanostructures.

Polymer-Based Nanostructures		*In Vivo* or *In Vitro* Test	Antibacterial Activity	Ref
Polymer	Nano Particles			
Chitosan, polyvinyl alcohol	Zinc oxide	Rabbits	*Escherichia coli*, *Pseudomonas aeruginosa*, *Bacillus subtilis*, *Staphylococcus aureus*	[195]
Chitosan and cellulose acetate	Cerium oxide	-	*Escherichia coli*, *Staphylococcus aureus*	[187]
Chitosan nanofibers	Graphene	Beveren rabbits	*Escherichia coli*, Agrobacterium, and yeast cells	[206]
Gelatin methacryloyl hydrogel	Cerium-containing bioactive glass	Sprague Dawley rats	*Escherichia coli*, *Staphylococcus aureus*	[207]
Hyaluronic acid	Zinc oxide	Human skin fibroblasts(CCD-986k)	*Staphylococcus aureus*, *Escherichia coli*	[208]
Sodium alginate, Poly (vinyl alcohol)	Graphene oxide	mouse embryonic fibroblasts (NIH 3T3 cells)	*Escherichia coli*, *Staphylococcus aureus*	[209]
Poly (ε caprolactone)/gelatin	Cerium oxide	Wistar rats/L929 murine fibroblastic cell line	-	[210]
Chitosan films	Silver	Human skin keratinocytes NCTC2544	*Pseudomonas aeruginosa*, *Staphylococcus epidermidis* and *Staphylococcus aureus*	[211]
Chitosan/silk sericin	Zinc oxide	Immortalized human keratinocyte cell line (HaCaT)	*Escherichia coli*, *Staphylococcus aureus*	[212]
Chitosan films	Curcumin/ silver	Albino rats	-	[213]

## Data Availability

Not applicable.

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
