# Peer review of "Polymer-Based Nanostructures for Pancreatic Beta-Cell Imaging and Non-Invasive Treatment of Diabetes"

_pharmaceutics, 2023, doi:10.3390/pharmaceutics15041215_

Round 1

Reviewer 1 Report

The purpose of this review was to discuss the use of nanostructures based on natural and synthetic polymers in the management of diabetes that may facilitate the treatment and diagnosis of this disease in three areas: drug delivery, imaging, and wound healing. This is highly relevant considering that diabetes has become one of the main non-communicable health problems of the 21st century, with most cases being found in developing countries, although recently also in developed countries, where the disease has spread. become the leading cause of premature death and disability. For this reason, new strategies are being sought that imply an improvement in the administration of drugs for the treatment of diabetes, the diagnosis and monitoring of this disease (beta cell mass), as well as the wound healing process (ulcers in feet and legs) from images obtained by techniques such as fluorescence and magnetic resonance.

Based on the foregoing, this review highlights the importance of analyzing the findings so far regarding the development, use and application of nanostructures (based on polymers), since this points towards a promising future. in the field of diabetes and its micro and macrovascular complications. For example, the importance of using formulations based on nanopolymers that are much more effective in releasing high concentrations of insulin that favor greater bioavailability of this hormone is mentioned. However, emphasis is also placed on the need to carry out more preclinical and clinical studies to determine the safety and efficacy of polymeric nanostructures, since the pharmacokinetics, biodistribution, toxicity and efficacy of the delivery systems may be essential to improve the results. patient results.

Therefore, this review article is very relevant and is very well designed, organized, and presented. It is only suggested to review some errors in the wording, since some terms are written in full and their respective abbreviations in parentheses, but later they become to write complete (eg beta cell mass), or viceversa, its abbreviation is put without having mentioned the complete term for the first time (MRI).

Author Response

attached file

Reviewer 2 Report

The opinion of this reviewer is that this article is well conceptualized, structured and highly impacting on scientific community, because it analyses correctly the use of polymer-based nanostructures in the management of diabetes.
Indeed, nanosciences can facilitate treatments and diagnosis in all medical fields, here summarized as imaging, drug delivery and wound healing.
All sections are detailed ad well described. But, as a personal point of view, some suggestions need to be provided to the authors, in order to complete, emphasize and exacerbate concepts which have been well described in any case.

1 - A hint at other innovative forms of diabetes treatment might be appropriate, such as nutritional and nutraceutical interventions.

Mare R. et al. - A new breakfast brioche containing bergamot fiber prevents insulin and glucose increase in healthy volunteers: a pilot study. - Minerva EndocrinologyVolume 46, Issue 2, Pages 214 - 2252021

2 - With regard to the medical applications of nanoparticles, it would be appropriate to mention a few examples, perhaps mentioning the active targeting obtainable by combining specific ligands, with appropriate references such as:

- vaccines

Rong-Rong Zhang et al. - Rational development of multicomponent mRNA vaccine candidates against mpox - Emerg Microbes Infect. 2023 Mar 22;2192815. doi: 10.1080/22221751.2023.2192815.

- anticancer

3 - Concerning insulin delivery a mention to microneedles should be appropriated.

Ling MH et al. - Dissolving polymer microneedle patches for rapid and efficient transdermal delivery of insulin to diabetic rats. Acta Biomater. 2013 Nov;9(11):8952-61. doi: 10.1016/j.actbio.2013.06.029. Epub 2013 Jun 29. PMID: 23816646”.

4- Last but not least, a slight revision of the English editing is recommended, in order to refine some sentences.

Author Response

attached file

Reviewer 3 Report

I suggest the addition of following items for improvement.

What are the problems associated with the use of polymers?

What are the major barriers and challenges is these approaches?

Author Response

attached file

Round 2

Reviewer 2 Report

Thank you for your kind answers.